# Laterality, sexual dimorphism, and human vagal projectome heterogeneity shape neuromodulation to vagus nerve stimulation

Natalia P. Biscola[1], Petra M. Bartmeyer[1], Youssef Beshay [2], Esther Stern[1], Plamen V. Mihaylov[3], Terry L. Powley[4], Matthew P. Ward [2,5] & Leif A. Havton [1,6,7] ✉

Neuromodulation by vagus nerve stimulation (VNS) provides therapeutic benefits in multiple medical conditions, including epilepsy and clinical depression, but underlying mechanisms of action are not well understood. Cervical vagus nerve biopsies were procured from transplant organ donors for high resolution light microscopy (LM) and transmission electron microscopy (TEM) to map the human fascicular and sub-fascicular organization. Cervical vagal segments show laterality with right sided dominance in fascicle numbers and cross-sectional areas as well as sexual dimorphism with female dominance in fascicle numbers. The novel and unprecedented detection of numerous small fascicles by high resolution LM and TEM expand the known fascicle size range and morphological diversity of the human vagus nerve. Ground truth TEM quantification of all myelinated and unmyelinated axons within individual nerve fascicles show marked sub-fascicular heterogeneity of nerve fiber numbers, size, and myelination. A heuristic action potential interpreter (HAPI) tool predicts VNS-evoked compound nerve action potentials (CNAPs) generated by myelinated and unmyelinated nerve fibers and validates functional dissimilarity between fascicles. Our findings of laterality, sexual dimorphism, and an expanded range of fascicle size heterogeneity provide mechanistic insights into the varied therapeutic responses and off-target effects to VNS and may guide new refinement strategies for neuromodulation.

Neuromodulation by vagus nerve stimulation (VNS) has shown therapeutic effects in multiple medical conditions. VNS was initially approved by the US Food and Drug Administration (FDA) for adjunctive treatment of medication-resistant epilepsy[1,2], treatment-resistant depression[3–5], and to augment rehabilitation of upper extremity function after an ischemic stroke[6]. Transcutaneous stimulation of the vagus nerve has also shown utility for the treatment of cluster headaches[7]. Emerging new indications for possible future VNS applications include inflammatory conditions[8–10]. However, underlying mechanisms of VNS action and physiology for extensive inter-individual variability in therapeutic outcomes are not well understood.

Therapeutic utility of VNS is limited by off-target effects. Surgical electrode placement for adjuvant VNS treatment is performed on the left-sided cervical vagus nerve segment[11] in attempts to reduce the risk for cardiac arrhythmias[12]. However, VNS titration for therapeutic effects is often limited by the early recruitment of somatic motor fibers within the laryngeal nerve projections to the larynx and vocal cords with hoarseness and swallowing difficulties[13–15]. New evidence-based strategies for refined VNS protocols are critically needed to optimize therapeutic VNS responses and limit off-target effects.

The anatomical and functional organization of vagal trunks is asymmetric across mammals, and ultrasound studies of the human cervical vagus have suggested a larger caliber on the right side[16]. The inferior laryngeal nerves show distinct asymmetry with the left-sided branch looping under the aortic arch, and the right-sided branch forming a loop under the right

[1]Department of Neurology, Icahn School of Medicine at Mount Sinai, New York, NY, USA. [2]Weldon School of Biomedical Engineering, Purdue University, West Lafayette, IN, USA. [3]Department of Surgery, Indiana University School of Medicine, Indianapolis, IN, USA. [4]Department of Psychological Sciences, Purdue University, West Lafayette, IN, USA. [5]Department of Medicine, Indiana University School of Medicine, Indianapolis, IN, USA. [6]Department of Neuroscience, Icahn School of Medicine at Mount Sinai, New York, NY, USA. [7]James J. Peters Department of Veterans Affairs Medical Center, Bronx, NY, USA. ✉e-mail: leif.havton@mssm.edu

subclavian artery[17]. Asymmetry in cardiac vagal innervation makes right-sided VNS more likely to cause bradycardia[18]. Abdominal organs also show differential innervation by the vagus nerve, as the left-sided trunk rotates to form the anterior abdominal vagal nerve to innervate the liver, spleen, and ventral stomach[19–21]. The right-sided trunk rotates to form the posterior abdominal vagal nerve, has no hepatic branch, and it innervates the celiac plexus, the dorsal stomach wall, colon, and uterus[22–24]. The vagal motor nuclei also show sexual dimorphism with regards to, for instance, androgen and estrogen receptor expressions[25]. In addition, GABA-ergic signaling in dorsal vagal motor neurons shows sexual dimorphism and fluctuates in females during the ovarian cycle[26].

Computational models and simulations to characterize electrical nerve thresholds have suggested multiple factors to influence nerve fiber recruitment by VNS, including fascicle cross-sectional area, perineurium thickness, nerve fiber caliber, and myelination, in addition to the stimulating electrode type and positioning, and distance between the stimulating electrode and its nerve fiber targets[27–29]. However, progress in VNS modeling and off-target effect attenuation are markedly limited by the sparsity of human ground truth data on the fascicular and sub-fascicular organization of the vagus nerve to guide the refinement of stimulation algorithms.

New protocols to procure and process vagus nerve tissues from transplant organ donors after brain death have allowed for unprecedented high resolution ultrastructural studies and detailed characterization of both myelinated and unmyelinated nerve fibers[30]. Here, laterality, sex as a biological variable, and heterogeneity of the vagal fascicular and sub-fascicular organization were identified as novel mechanistic contributors to the diversity of evoked VNS responses, inclusive of both therapeutic and off-target effects. We also introduced a new heuristic action potential interpreter (HAPI) tool and validated ground truth data-driven predictions for human vagal nerve fiber recruitment to VNS.

## Results

Cervical vagus nerve biopsies were procured from male and female organ donors. The age of the male donors was $44.3 \pm 3.7$ years ($n = 14$), and the age of the female donors was $48.2 \pm 3.1$ years ($n = 13$). There was no statistical difference between the male and female study groups.

### Laterality of the cervical vagus nerve

The distal cervical trunk segment of the human vagus nerve showed multiple fascicles separated by a loose epineurium in plastic resin-embedded sections stained with toluidine blue and viewed by light microscopy (LM) (Fig. 1A). The fascicles varied in size and shape, and some fascicles included subfascicles separated by perineurium borders (Fig. 1B, C). The number of fascicles in the cervical trunks of the right-sided vagus nerve was $11.4 \pm 2.0$ ($n = 21$) and significantly larger than the corresponding left-sided fascicle number of $5.5 \pm 1.4$ ($n = 15$) ($p < 0.05$) (Fig. 1D). Quantitative analysis of fascicle size showed a significantly larger total fascicle area on the right side at $820 \pm 97$ ($\times 1000\ \mu m^2$) ($n = 21$) compared to the left side at $583 \pm 97$ ($\times 1000\ \mu m^2$) ($n = 15$) ($p < 0.05$) (Fig. 1E). The total endoneurium area was also significantly larger on the right side at $691 \pm 84$ ($\times 1000\ \mu m^2$) ($n = 20$) compared to the left side at $492 \pm 82$ ($\times 1000\ \mu m^2$) ($n = 20$) ($p < 0.05$) (Fig. 1F). Paired analyses of the right and left cervical vagal samples procured from the same donor provided additional support for an organizational asymmetry at the level of the cervical vagus nerve with a significantly higher number of fascicles on the right side ($n = 12$ pairs, $p < 0.01$) (Fig. 1G), a significantly larger total fascicle area on the right side ($n = 12$ pairs; $p < 0.001$) (Fig. 1H), and a significantly larger total endoneurium area on the right side ($n = 12$ pairs, $p < 0.001$) (Fig. 1I). Collectively, these quantitative studies showed laterality with a right sided dominance.

### Sexual dimorphism of the cervical vagus nerve

Analysis of fascicle organization and size was next performed on the cervical trunks from male and female donors as separate groups. Here, women showed a significantly higher total number of fascicles in a mixed population of left- and right-sided cervical vagal trunks at $13.1 \pm 2.5$ ($n = 15$) compared to men at $6.0 \pm 1.3$ ($n = 21$) ($p < 0.05$) (Fig. 1J). Taking into account the potential impact of laterality, the left and right sides were also analyzed separately. A significantly higher left-sided number of fascicles was seen in women at $8.6 \pm 2.4$ ($n = 7$) compared to men at $2.9 \pm 1.2$ ($n = 8$) ($p < 0.05$) (Fig. 1K). A significantly higher number of right-sided fascicles was also seen in women at $17.1 \pm 3.8$ ($n = 8$) compared to the corresponding number for men at $7.9 \pm 1.7$ ($n = 13$) ($p < 0.05$) (Fig. 1L). There was no statistical difference between men and women for the combined, left-sided, or right-sided total fascicle areas or endoneurium areas. The findings support sexual dimorphism with a right sided dominance for the number of cervical vagal fascicles in women.

### Fascicle heterogeneity of the cervical vagus nerve

Heterogeneity of individual fascicle shape and size was assessed for the bilateral cervical vagus nerve trunks. Circularity was determined by five separate methods, including the shape factor, form factor, aspect ratio, compactness, and roundness[31]. A circle has a shape factor of 3.54 and a value of 1 for the other four shape outcome measures. The shape factor, form factor, aspect ratio, compactness, and roundness were calculated for all left-sided fascicles as $4.56 \pm 0.06$, $0.63 \pm 0.02$, $0.62 \pm 0.01$, $0.72 \pm 0.01$, and $0.52 \pm 0.01$, respectively, ($n = 81$), and they were not statistically different from the corresponding right-sided shape measures of $4.48 \pm 0.03$, $0.65 \pm 0.01$, $0.65 \pm 0.01$, $0.74 \pm 0.01$, and $0.56 \pm 0.01$ ($n = 236$). Collectively, the studies indicated that the individual fascicles have a non-circular shape. Additional sub-group analysis of left- and right sided fascicles in men and women also indicated a non-circular fascicle shape but no sex-dependent statistical difference in shape between samples.

Next, a total of 317 individual fascicles underwent detailed size characterizations but without the assumption of a circular shape. The mean circumference was shorter for right-sided fascicles at $0.93 \pm 0.06$ (x1000 µm) ($n = 236$) compared to corresponding left-sided fascicles at $1.17 \pm 0.11$ ($n = 81$) ($n < 0.02$) (Fig. 2A). Both the total fascicle area and endoneurium area were smaller on the right side at $75.9 \pm 10.1$ (X1000 µm²) ($n = 236$) and $63.9 \pm 8.5$ (x1000 µm²) ($n = 236$) compared to the corresponding areas on the left side at $108.0 \pm 22.8$ (x1000 µm²) ($n = 81$) and $91.1 \pm 19.1$ (x1000 µm²) ($n = 81$) ($p < 0.05$), respectively (Fig. 2B,C). The mean circumference, total fascicle area, and endoneurium area of the left-sided fascicles in women ($n = 58$) and men ($n = 23$) were not statistically different. For the right-sided fascicles, the mean circumference, total fascicle area, and endoneurium area in women ($n = 134$) and men ($n = 102$) were also not statistically different. Therefore, subsequent analyses of individual cervical vagal fascicles included separate analysis of the right- and left-sided fascicles but did not separate fascicles based on sex as a biological variable.

### Size correlations of cervical vagus nerve fascicles

There was a strong linear correlation between endoneurium area and circumference with a coefficient of determination ($R^2$) of 0.87 for left-sided and 0.81 for right-sided cervical fascicles (Fig. 2D, E). There was also a strong linear correlation between the fascicle area and perineurium area with an $R^2$ of 0.85 for left-sided and 0.82 for right-sided cervical fascicles (Fig. 2F, G). In contrast, there was a weak correlation between fascicle area and the ratio of the endoneurium/fascicle areas with an $R^2$ of 0.02 for left-sided and 0.07 for right-sided fascicles (Fig. 2H, I), indicating that the data variability was not well explained by the linear regression model. However, correlational studies between log10 fascicle area and the ratio of endoneurium area/fascicle area showed a moderate correlation for both the left- and right-sided cervical fascicles (Fig. 2J). The ratio of endoneurium area/fascicle area was significantly higher at $0.81 \pm 0.02$ on the left side ($n = 81$) compared to $0.76 \pm 0.01$ on the right side ($n = 236$) ($p < 0.001$) (Fig. 2K), possibly influenced by a higher number of small fascicles on the right side with a relatively thicker perineurium compared to size. There was no difference between the left and right sides with regards to the ratio of endoneurium area/fascicle area in men or women.

### Sub-fascicular heterogeneity of the cervical vagus nerve

To determine the sub-fascicular organization of the cervical segment of the human vagus nerve, ultrastructural studies were performed to identify and

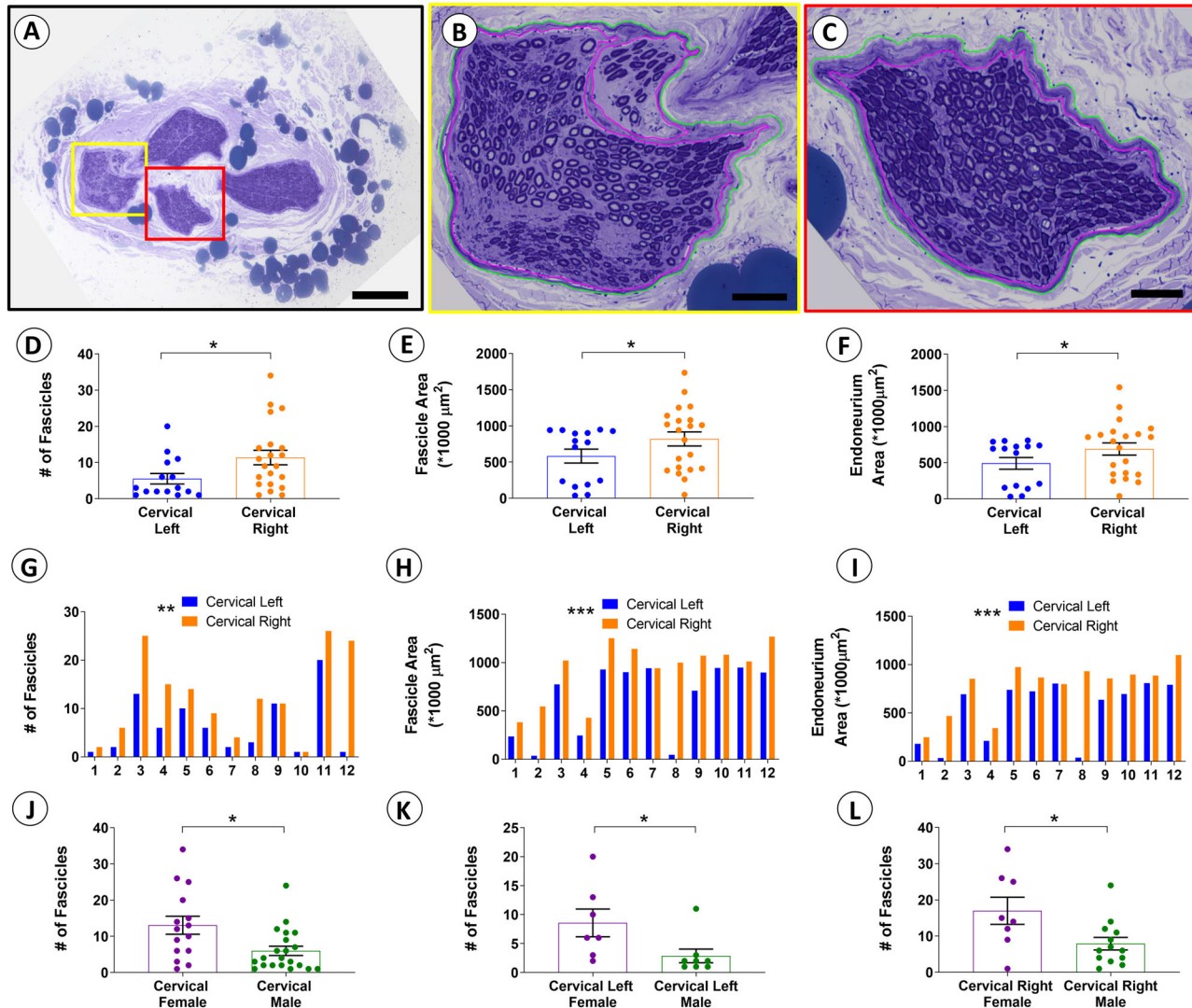

**Fig. 1 | The human cervical vagus nerve shows laterality and sexual dimorphism.** The human cervical vagus nerve shows multiple fascicles of varied sizes and shapes, and they may have sub-fascicles (**A–C**). Yellow and red boxed areas in **A** are at higher magnification in **B** and **C**. The perineurium is outlined in green and endoneurium in purple colors (**B**, **C**). The right side has significantly more fascicles at 11.4 ± 2.0 ($n$ = 21) compared to the left side at 5.5 ± 1.4 ($n$ = 15) ($p < 0.05$) (**D**). The right side shows a significantly larger total fascicle area at 820 ± 97 (×1000 μm$^2$) ($n$ = 21) compared to the left side at 583 ± 97 (×1000 μm$^2$) ($n$ = 15) ($p < 0.05$) (**E**). The total endoneurium area is also significantly larger on the right side at 691 ± 84 (×1000 μm$^2$) ($n$ = 20) compared to the left side at 492 ± 82 (×1000 μm$^2$) ($n$ = 20) ($p < 0.05$) (**F**). Paired analyses with right and left cervical vagal samples procured from the same donor also provides strong support for laterality at the level of the cervical vagus nerve with a significantly higher number of fascicles on the right side ($n$ = 12, $p < 0.01$) (**G**), significantly larger fascicle area on the right side ($n$ = 12 pairs; $p < 0.001$) (**H**), and significantly larger endoneurium space on the right side ($n$ = 12 pairs, $p < 0.001$) (**I**). Statistical analysis of fascicle-level data in both sexes showed a significantly higher total number of fascicles in women for a mixed population of left- and right-sided cervical fascicles at 13.1 ± 2.5 ($n$ = 15) compared to men at 6.0 ± 1.3 ($n$ = 21) ($p < 0.05$) (**J**). A significantly higher left-sided number of fascicles was seen in women at 8.6 ± 2.4 ($n$ = 7) compared to men at 2.9 ± 1.2 ($n$ = 8) ($p < 0.05$) (**K**). A significantly higher number of right-sided fascicles was detected in women at 17.1 ± 3.8 ($n$ = 8) compared to men at 7.9 ± 1.7 ($n$ = 13) ($p < 0.05$) (**L**).

characterize the size, myelination, and intra-fascicular location for all individual myelinated and unmyelinated nerve fibers. TEM montages of entire cervical fascicles ($n$ = 8) were constructed and followed by digital segmentation of all individual nerve fibers in each fascicle (Fig. 3). The fascicles (F1-F8) showed extensive heterogeneity in nerve fiber size and myelination, and the vast majority of nerve fibers in the present sample of fascicles were unmyelinated (Fig. 3). The number of myelinated fibers ranged from 4 to 3311 and the corresponding number of unmyelinated fibers ranged from 21 to 15,646 in the present sample of cervical fascicles ($n$ = 8) (Fig. 3). Size distribution analysis revealed extensive variability in the caliber of myelinated fibers and the degree of myelination by myelin thickness and G-ratio determinations, both within and between fascicles. G-ratio calculations were performed as axon diameter/myelinated fiber diameter. Size distribution analysis suggested both unimodal and bi- or multi-modal distribution

patterns for myelinated nerve fibers with diameter peaks within 1.5–2.5 μm and 8–11 μm ranges (Fig. 3), suggestive of a functional diversity of nerve fiber conduction and recruitment properties within the cervical vagus segments. Regardless of size, all cervical vagal fascicles shared multiple ultrastructural features, including a well-defined perineurium and an endoneurium space featuring both myelinated and unmyelinated fibers, Schwann cell nuclei, and collagen fibers (Fig. 4A–N). The small-caliber axons of unmyelinated fibers were ensheathed by non-myelinating Schwann cells in Remak bundles (Fig. 4D,G–J). Additional sub-fascicular comparisons of nerve fiber size and myelination between fascicles ($n$ = 8) showed significant differences in fiber and axon diameters, myelin thickness, and G-ratio for myelinated fibers between fascicles in multiple paired comparisons (Fig. 4O). Size comparisons between unmyelinated fibers also showed significant

**Fig. 2 | Individual human cervical vagal fascicles show laterality with left-sided dominance.** In side comparisons, the circumference of individual fascicles was significantly longer on the left compared to the right side ($p < 0.05$) (**A**), and the total fascicle and endoneurium cross-sectional areas were significantly larger on the left compared to the right side ($p < 0.05$) (**B**, **C**). Hence, the analysis of individual fascicles was performed separately for left- and right-sided fascicles. Correlational studies between cervical fascicle endoneurium area and circumference as well as between cervical fascicle and perineurium areas were next performed. There was a strong linear positive correlation between the endoneurium area and circumference on both the left side ($R^2 = 0.8668$) and the right side ($R^2 = 0.8076$) (**D**, **E**). There was also a strong linear positive correlation between the fascicle area and perineurium area on the left side ($R^2 = 0.8471$) and the right side ($R^2 = 0.8245$) (**F**, **G**). In contrast, the linear correlation was weak between the fascicle area and the ratio of the endoneurium area/fascicle area on the left side ($R^2 = 0.0153$) and the right side ($R^2 = 0.0727$) (**H**, **I**). Here the ratio of the endoneurium and fascicle areas was used as an indicator for the relative proportion of fascicle insulation by the perineurium with a lower number indicating a more prominent perineurium thickness for fascicle size. The correlational analysis hence indicated that the relationship between relative perineurium thickness and fascicle size is not linear. Close observation of the fascicle area distribution suggested an uneven spread of the data points with a majority of the fascicles clustering within the relatively smaller size range on both the left and right sides (**D–I**). Next, correlational studies between log10 fascicle area and the ratio of endoneurium area/fascicle area showed a strong correlation on both the left and right side (**J**). Additional analysis showed the ratio of endoneurium area/fascicle area as significantly higher for left- compared to right-sided fascicles ($p < 0.001$) (**K**). The latter finding was influenced by larger left-sided fascicles (see **A–C**) with also a thinner perineurium for fascicle size and associated higher endoneurium area/fascicle area ratio. $R^2$ = Coefficient of determination. Red dotted lines in **D–I** indicate 95% confidence intervals.

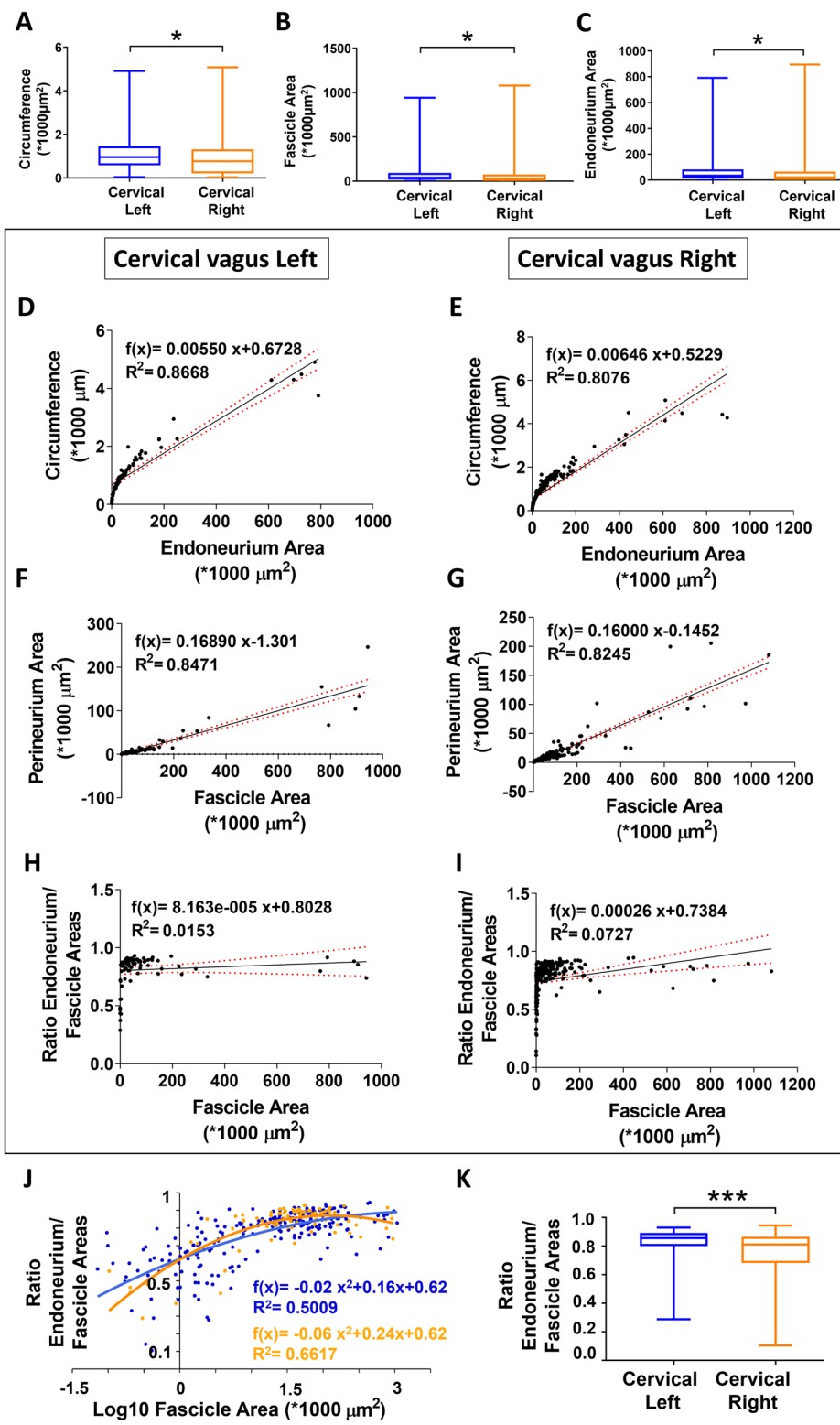

differences between multiple fascicles in paired comparisons (Fig. 4O). Collectively, these quantitative studies suggest different functional phenotypes between fascicles with regards to conductive properties, recruitment, and potential innervation targets.

**Evoked response predictions to vagus nerve stimulation**
Human studies on the recruitment and conductive properties of vagal nerve fibers to electrical stimulation are challenging, especially for the smallest

myelinated and unmyelinated fibers with high activation thresholds. Hence, we introduce a heuristic action potential interpreter (HAPI) tool to simulate anticipated responses to VNS (Fig. 5). The HAPI takes into consideration multiple TEM measures of individually segmented myelinated and unmyelinated nerve fibers to predict evoked compound nerve action potentials (CNAPs) (Fig. 5). Heterogeneity between fascicles of the cervical vagus include variability with regards to fascicle size and nerve fiber compositions (Fig. 3), which will influence the predicted CNAP responses to

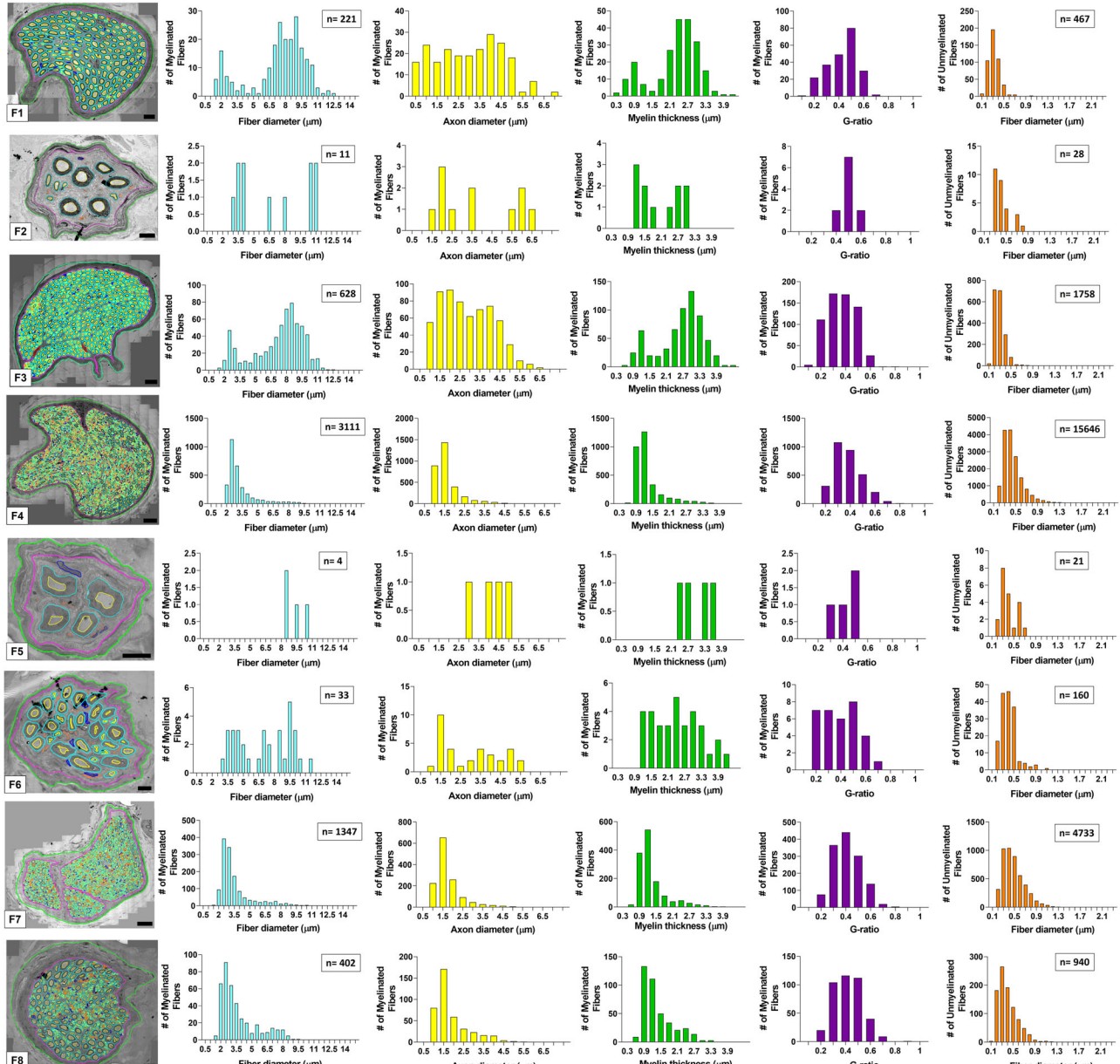

**Fig. 3 | Ultrastructure of the human cervical vagal sub-fascicular organization.** All myelinated and unmyelinated axons of eight representative fascicles (F1-F8) were digitally segmented for quantitative analysis of myelinated and unmyelinated nerve fiber numbers, diameter of myelinated fibers, axonal diameters (without myelin), myelin thickness, and G-ratio (axon diameter/myelinated fiber diameter). The total number of myelinated unmyelinated fibers are indicated in separate boxes for each fascicle. Note extensive variability between fascicles in nerve fiber numbers, relative compositions of myelinated and unmyelinated fibers, myelination patterns (myelin thickness and G-ratio), and nerve fiber size distributions. Unimodal distributions with predominantly small myelinated fibers and bi- or multi-modal distributions with both small and larger fiber distribution peaks are represented. The much varied sub-fascicular organization of nerve fibers suggests extensive fascicular heterogeneity for nerve conduction and recruitment properties. Scale bar = 20 μm in F1, 10 μm in F2, 30 μm in F3, 40 μm in F4, 10 μm in F5 and F6, 40 μm in F7, and 20 μm in F8.

electrical stimulation. Smaller fascicles with fewer myelinated and unmyelinated nerve fibers were predicted to produce CNAPs with smaller amplitude compared to larger fascicles, and latencies to CNAP peaks depended on nerve fiber diameter and myelination (Fig. 6). In the present sample, broad size ranges of myelinated fibers were shown with predicted CNAP latencies within the ranges for group A and B fibers, suggestive of the possible inclusion of general visceral afferents, preganglionic parasympathetic fibers, and somatic motor fibers (Fig. 6). A distinct group of fibers with predicted latencies within the C-fiber range of unmyelinated fibers was also present in both small and larger fascicles (Fig. 6). Distinct sub-classes of vagal fibers, based on functional properties or ultrastructural features, may be unevenly distributed across the cross-sectional space. The relative contributions of such sub-sets of nerve fibers to the overall predicted

CNAP may be determined based on their axonal diameter, myelination, and clustered centroid locations within a fascicle (Fig. 7).

Some fascicles showed division of the endoneurium space into two or more sub-fascicles, separated by extensions of the perineurium. A total of 49 of the 317 cervical fascicles (15.5%) in the present sample showed sub-fascicular compartments. Sub-fascicle compartments were readily identified in left- and right-sided fascicles from both men and women. However, Pearson's chi-squared analysis did not show any significant signs of laterality or sexual dimorphism with regards to the frequency of sub-fascicle presence in the cervical vagus nerve. For the purpose of neuromodulation, sub-fascicles of the cervical vagus nerve may be regarded as distinct functional units. They are characteristically populated by both myelinated and unmyelinated fibers (Fig. 8). Predicted CNAP responses to VNS may be

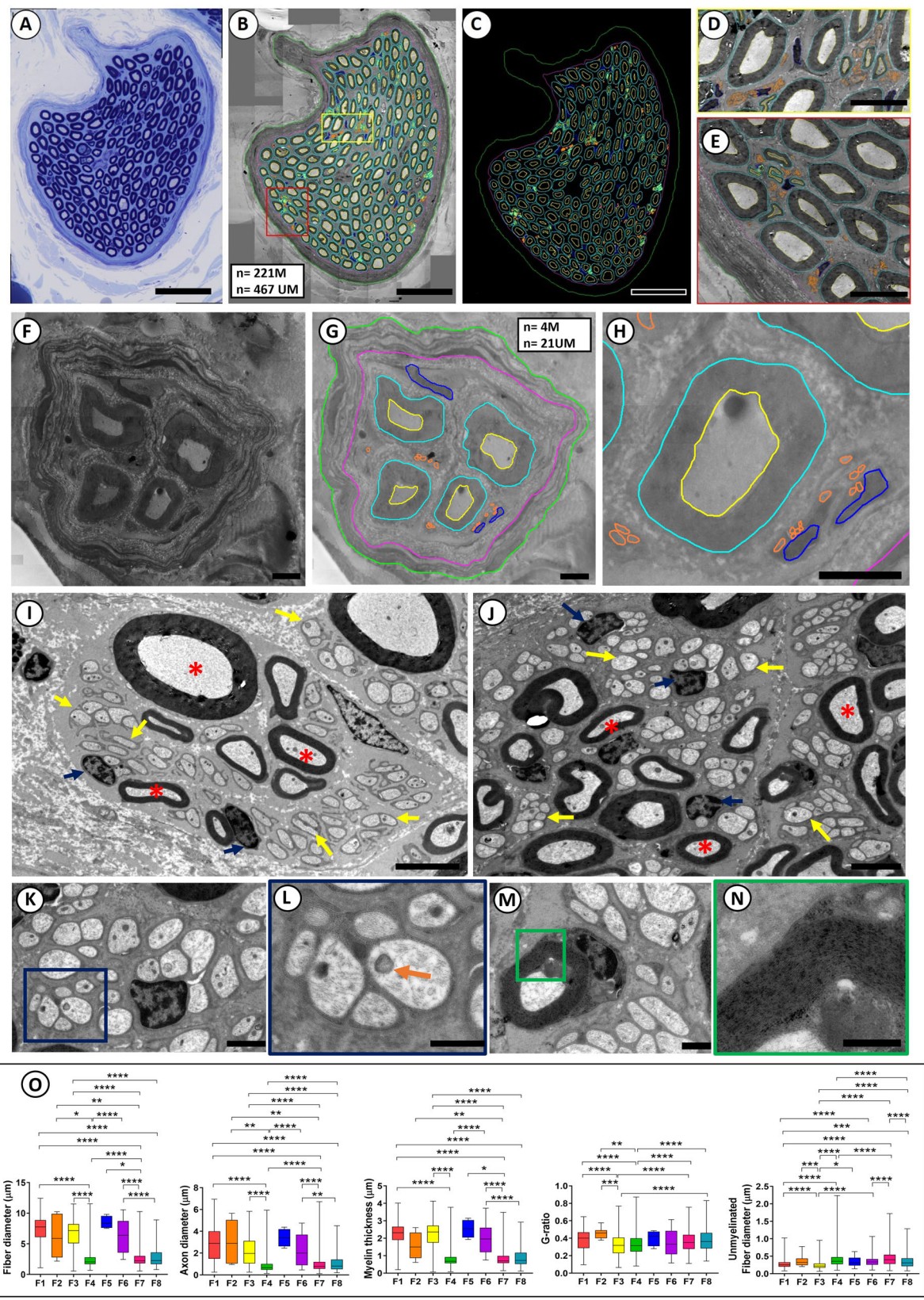

determined for each individual sub-fascicle and for the fascicle as a whole (Fig. 9). The HAPI predictions of CNAP waveform and peak amplitudes reflect the composition and size features of both myelinated and unmyelinated axons (Fig. 9). This important property of the HAPI tool allows for a more systematic method of decoding and classifying the organization and functions of the vagus nerve. Knowledge of the location, morphology, and extracellular signaling characteristics, along with the functional properties of the individual nerve fibers, will be essential to inform future bioelectronic medical therapies that target specific fibers or regions of a nerve using, e.g., nerve-response feedback and intelligent micro-stimulation algorithms.

**Fig. 4 | Ultrastructural heterogeneity of human cervical vagal fascicles.** LM micrograph shows toluidine blue-stained, medium sized, fascicle of human cervical vagus nerve segment (**A**). TEM montage of fascicle in **A** is shown with super-imposed digital segmentation of 221 myelinated (M) and 467 unmyelinated (UM) axons (**B**). Outer and inner contours of myelin sheets are shown in blue and yellow, and UM axons in orange (**B**). Outer and inner contours of the perineurium are presented in green and purple, and Schwann cell (SC) nuclei are shown in blue. Segmentations without TEM montage are shown in **C**. Yellow and red boxed areas in **B** are shown at high magnification in **D** and **E**. Note clusters of small M axons, UM axons, and SC nuclei in between larger M fibers. TEM image of a small fascicle is shown in **F** and with superimposed digital segmentation of outer and inner contours of the perineurium and M and UM fibers, and SC nuclei (**G**). The color coding in **G** is identical to **B**–**E**. Higher magnification TEM detail of lower right portion of **G** is shown in **H**. TEM of representative areas of cervical vagus nerve segment shows M

and UM fibers, and SC nuclei indicated by red asterisks, yellow arrows, and blue arrows, respectively (**I**, **J**). TEM shows detail of fascicle endoneurium with UM axons, Schwann cell nucleus, and surrounding collagen fibers (**K**). Higher magnification of blue box area in **K** shows UM fibers with intra-axonal cytoskeletal detail (**L**). An intra-axonal mitochondria is shown with orange arrow. An M axon and associated myelinating SC is shown in **M**. High magnification of green box in **M** identifies individual wraps of myelin sheet (**N**). Analysis of all segmented M and UM fibers in fascicles F1-F8 (Fig. 3) compares fiber size and myelination between fascicles (**O**). Paired statistical analysis shows significant differences in mean M fiber and axonal diameters, myelin thickness, and G-ratio between multiple fascicle pairs (**O**). The mean diameter for UM fibers also shows a statistical difference between many fascicles in paired analyses (**O**). Scale bar = 50 μm in **A**–**C**, 10 μm in **D**, **E**, 5 μm in **F**, **G**, 15 μm in **H**, 5 μm in **I**, **J**, 2 μm in **K**, 1 μm in **L**, **M**, and 0.5 μm in **N**.

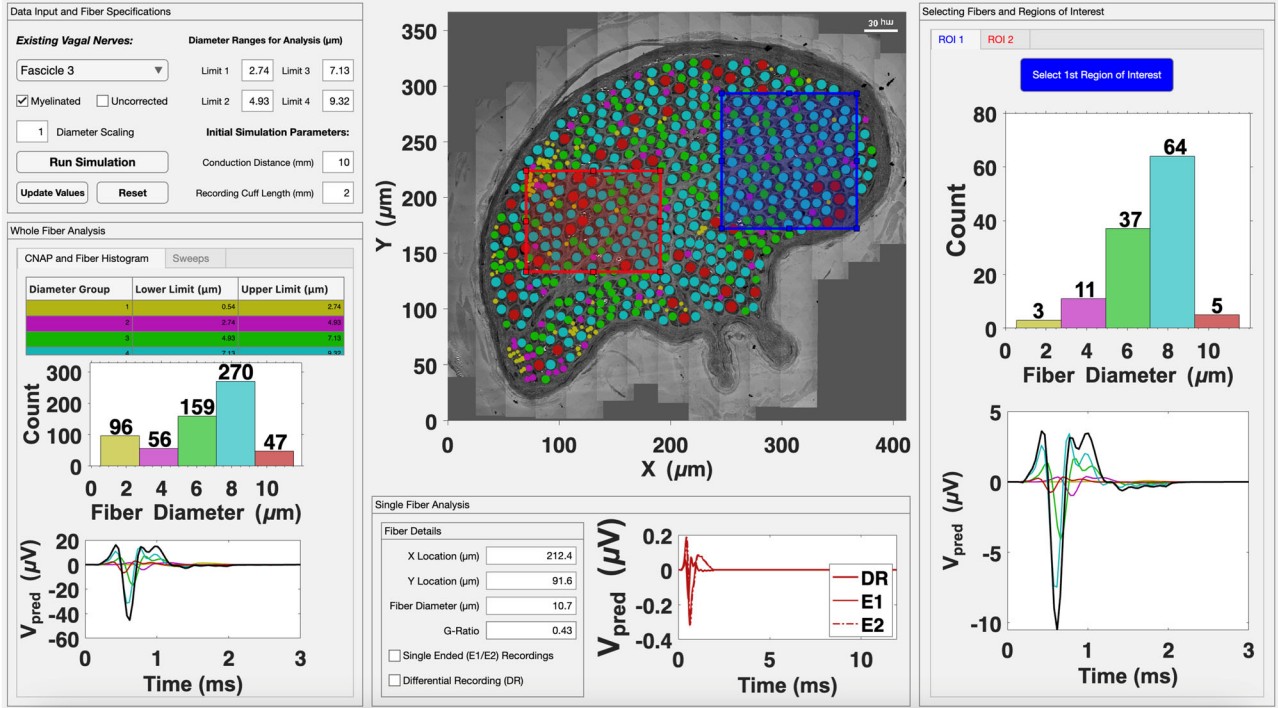

**Fig. 5 | Heuristic Action Potential Interpreter (HAPI) – an open-source tool to explore relationships between nerve composition/organization, fiber morphology, conduction properties, and expected CNAP responses.** Screenshot shows a custom analysis of the organization of a fascicle from a human vagus nerve. HAPI allows users to load in segmentation data and select up to five colors to represent different fiber/axon diameter ranges, and bin limits can be automatically optimized or manually assigned. Users can select any fiber and the software will show its morphometry data and the predicted shape and latency for the corresponding single fiber action potential at a chosen conduction distance/recording configuration (e.g.,

differential versus single ended recordings). Users can simulate CNAPs by selecting different nerve or fascicle regions, the full nerve, or a particular sub-set of fibers within a range of diameters. Regions can be compared in terms of organization/ composition and expected CNAP response. Users can also produce a sweep of CNAP responses across a wide range of conduction distances and recording cuff lengths, which refers to the distance between the two electrodes in a cuff used to record the CNAP. Colors allows for easy visualization of where and how specific fiber groups contribute to the bulk CNAP response.

Combined analysis of all myelinated and unmyelinated fibers allows for predictions of the maximal CNAP response for individual vagal fascicles (Fig. 10). We note that the HAPI-predicted CNAP response is highly dependent on the simulated recording conditions, including the conduction distance or recording electrode separation, and even slight changes in these conditions would influence, for instance CNAP latencies. Here, the predicted CNAP response versus conduction velocity shows CNAP volleys with varied conduction speeds, including representation of vagal nerve fibers in the Aγ, Aδ, B, and C fiber ranges (Fig. 10).

## Discussion
Fascicle laterality, sexual dimorphism, and heterogeneity were identified here as novel and key organizational features of the human vagus nerve in cervical segments procured from transplant organ donors. Our combined

LM and TEM approach for detailed morphological mapping of the cervical vagal segments provided inclusion of all individual fascicles, including numerous small fascicles requiring high resolution LM and ultrastructural studies for their first-time detection and validation. The small vagal fascicles were readily identified in both males and females, and they contributed to a right-sided dominance in both men and women, sexual dimorphism with a higher number of cervical vagal fascicles in women, and heterogeneity with regards to marked inter-fascicle variability in the composition of myelinated and unmyelinated nerve fibers. The ultrastructural mapping of all individual myelinated and unmyelinated fibers in single vagal fascicles provided ground-truth segmented data for realistic predictions of a broad range of evoked CNAP responses and recruitment to simulated VNS.

An unexpected finding in the present study was the demonstration by high-resolution microscopy of a large number of small fascicles, many of

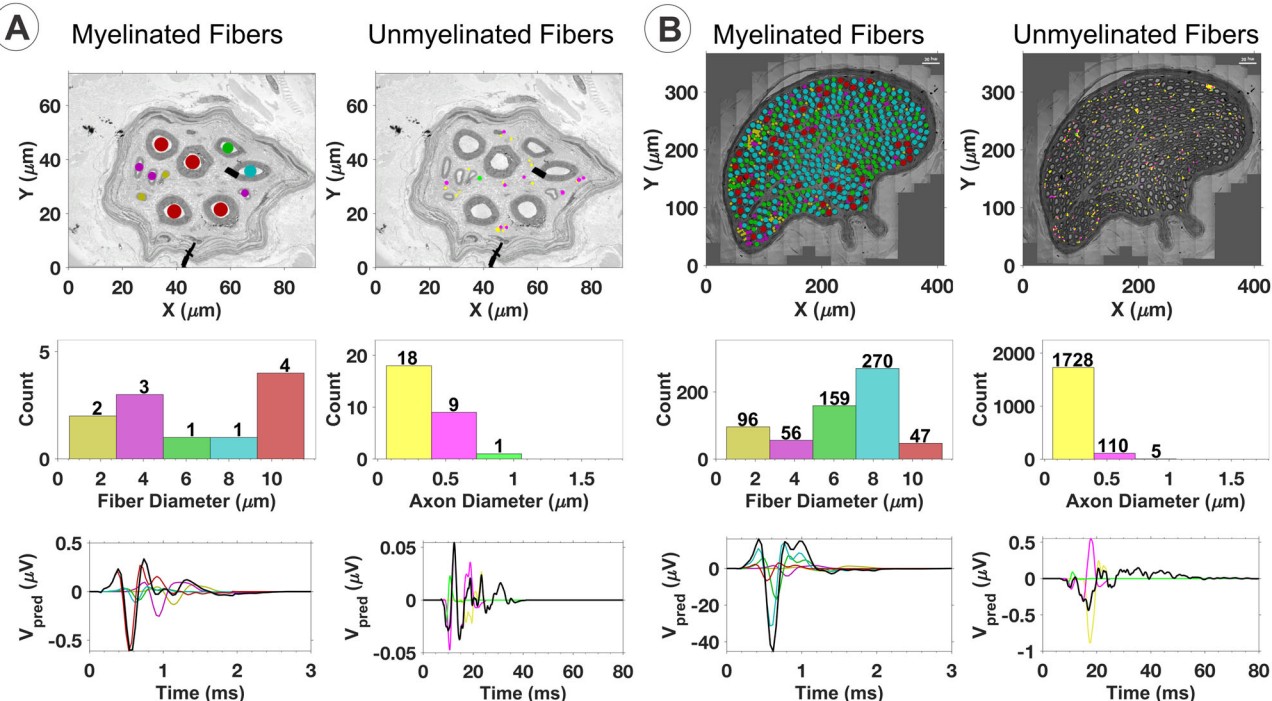

**Fig. 6 | Fascicle size, composition, and electrode configuration determine the overall shape and features of the CNAP predictions by HAPI.** Multiple fascicle sizes are present in the human vagus, each with their own unique fiber composition and organization. In **A**, we show by TEM an example of a small fascicle with 11 myelinated fibers (left column) and 28 unmyelinated fibers (right column). In **B**, we show an example of a larger fascicle with substantially more myelinated and unmyelinated fibers. Note color coding of individual fiber location, diameter and corresponding contributions to the CNAP response from myelinated and unmyelinated fibers in **A** and **B** for a conduction distance of 10 mm using a bipolar platinum/iridium (Pt/Ir) recording cuff electrode with 0.5 mm inner diameter and 2.0 mm inter-electrode spacing. The relative magnitude of features of the predicted CNAP for the larger fascicle is much greater than those of the small fascicle CNAP, reflecting contributions from all myelinated and unmyelinated axons.

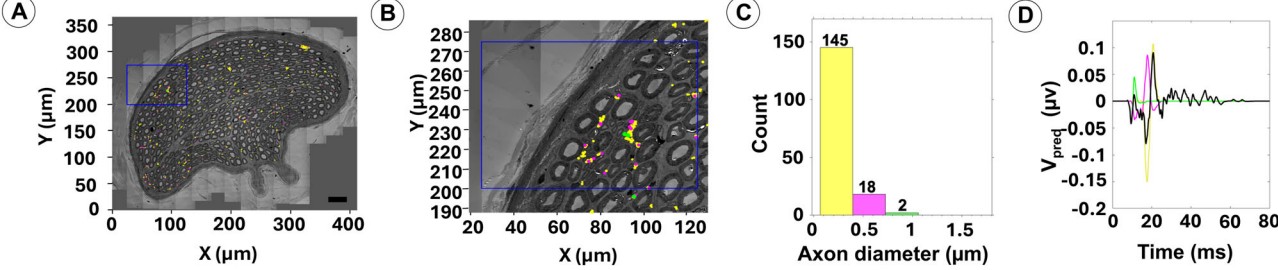

**Fig. 7 | The HAPI tool predicts CNAP responses from unmyelinated axons of the human vagus nerve.** An overview of TEM of fascicle 3 in Fig. 3 is shown in **A** with area indicated by blue box displayed at higher magnification and with segmented unmyelinated axons superimposed in **B**. Note clusters of unmyelinated fibers in between larger myelinated axons. Histogram of unmyelinated fiber size distribution is shown in **C**. The HAPI tool provides CNAP predictions with color-coded display of the relative contributions from unmyelinated axons of different sizes in **D**. The CNAP predictions here are based on a recording distance of 10 mm and the use of a bipolar Pt/Ir recording cuff electrode with an inner diameter of 0.5 mm and 2.0 mm inter-electrode spacing.

which were markedly below the size range identified by ultrasound morphometry[32], micro-CT imaging[33,34], and LM of cadaver-derived tissues after Masson trichrome or hematoxylin & eosin histochemical staining[35–37]. In contrast to prior morphological studies of frozen or paraffin-embedded cadaveric tissues, human vagus nerve segments were here procured from transplant organ donors and next processed and embedded in plastic resin for high resolution LM and TEM studies[30]. Our approach allowed for the identification of vagal fascicles across the full size spectrum and detailed sub-fascicular characterization of the endoneurium, including morphological mapping of individual myelinated and unmyelinated nerve fibers[30]. In the present study, the first quartile of cervical vagal fascicles had a circumference of less than 300 μm, and several fascicles within this size range may have escaped detection in earlier studies due to the technical limitations of the chosen methods for non-invasive imaging or many standard histological preparations. Interestingly, recent finite element modeling of human vagus nerve stimulation have suggested that fascicles with a thinner perineurium and smaller cross-sectional area provide lower nerve fiber activation thresholds to electrical simulation with a cuff electrode[29]. Our identification of a novel group of small vagal fascicles with a thin perineurium within the cervical trunk of the human vagus nerve is therefore of significant relevance for clinical neuromodulation. Provided that all other variables remain constant, nerve fibers within the smallest vagal fascicles represent the most likely candidates for early recruitment during VNS. Expanded ultra-structural characterization of small fascicles of the human vagus, including mapping of their myelinated and unmyelinated axons, will provide critical ground truth data for refined modeling studies and a unique opportunity to

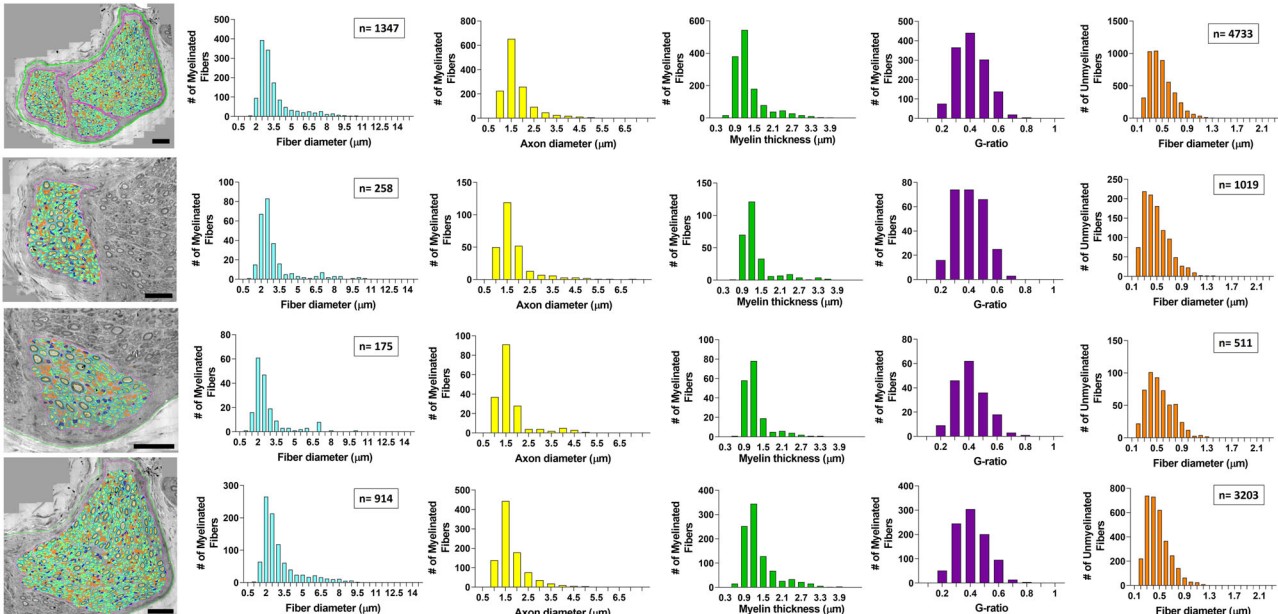

**Fig. 8 | Fiber composition of vagal sub-fascicles.** The myelinated and unmyelinated axons of each individual sub-fascicle of Fascicle 7 in Fig. 3 were segmented to provide nerve fiber size distributions. The myelinated fiber diameter and corresponding diameter of the axon proper (without myelin), myelin thickness, G-ratio, and diameter of unmyelinated axons are presented for each sub-fascicle and for the whole fascicle. Scale bars = 40 μm.

gain insights into potential mechanisms underlying neuromodulation by electrical stimulation of the human vagus nerve. Future studies may also be designed to determine whether sex is also a biological variable for the sub-fascicular organization of the cervical vagus nerve. If so, such findings may influence additional refinement of VNS treatment strategies to reverse conditions in both men and women.

Laterality with a right-sided dominance was demonstrated in the present study with regards to fascicle numbers as well as total cross-sectional fascicle and endoneurium areas in both men and women as well as when the data for the two sexes were combined. A recent human cadaveric study reported a larger cross-sectional area for the right-sided cervical vagus nerve after casting of the nerve segment and histology[38], and a single study has proposed more cervical segment fascicles on the right side but did not demonstrate any side difference in fascicle cross-sectional area[39]. However, most anatomical studies did not detect any side difference in fascicular numbers or size measurements[35,37,40]. Differences in study design and methods used may have contributed to the varied outcomes. First, the present study used organ donor-procured vagus nerve tissues, instead of post-mortem tissues from autopsies, to minimize autolysis of nervous tissues. Second, plastic embedding of nervous tissues allowed for high resolution microscopy, including identification of vagal fascicles across the full size range as well as all individual myelinated and unmyelinated axons for ground truth data collection by LM and TEM studies. Third, procurement of bilateral cervical vagus nerve samples form organ donors allowed for paired statistical analysis. Future ultrastructural studies will be needed to determine whether laterality may also be present at the sub-fascicular level with regards to the distribution of various nerve fiber populations based on fiber size, myelination, and conduction properties.

Electrical stimulation of the vagus nerve may cause membrane depolarization of axons and initiate the propagation of action potentials, but the threshold for individual nerve fiber recruitment and conduction properties in the peripheral nervous system are influenced by multiple morphological factors, including fiber size, myelination, and internode distance[41]. The human vagus nerve is a mixed nerve and its projectome includes motor, sensory, and autonomic fibers of much varied sizes, myelination, and conduction velocities[30,42]. The myelinated motor axons form group A fibers and may be sub-divided by size criteria and conduction properties into Aα, Aβ, Aγ, and Aδ fibers[43–46]. The Aα- and Aγ-fibers innervate extra- and intra-fusal skeletal muscle fibers, respectively, whereas Aβ-fibers innervate both extra- and intra-fusal skeletal muscle fibers[47]. The smallest efferents form thinly myelinated Aδ-fibers and include preganglionic autonomic fibers[31]. Myelinated sensory fibers, regardless of muscle, cutaneous, or autonomic origin, are similarly divided based on size and conduction properties into groups I, II, and III with group I fibers subdivided into groups Ia and Ib for muscle spindle and tendon organ afferents, respectively, and unmyelinated afferent fibers distinguished as a separate group IV[46,48–50]. Here, our sub-fascicular TEM mapping of myelinated and unmyelinated fibers of the human vagus nerve in combination with HAPI predictions of corresponding CNAPs showed inclusion of human vagus nerve fibers across the full size and conductive property ranges of group A and B motor fibers, group C fibers, and groups I-IV afferents.

Future strategies for modulating the nervous activity of vagal segments by electrical stimulation may consider, for instance, invasive and non-invasive strategies, use of sensory feed-back loops, energy-efficiency, and multi-channel electrode arrays[51–55]. Improved electrode designs and the introduction of customized stimulation protocols may help to optimize treatments and reduce off-target effects, thereby achieving critical goals of precision medicine and personalized therapeutics[56]. For this purpose, the HAPI tool may guide the development of new and refined neuromodulation strategies by providing an interactive platform for the use of segmented ground truth data on nerve fiber size to determine the influence of a variety of distinct measurement conditions, including electrode dimensions, stimulation parameters, conduction distance, and interelectrode spacing for realistic predictions of evoked CNAPs. Such information may help guide the development of refined intra-operative monitoring and decision-making during VNS electrode placement.

In conclusion, the present high-resolution LM and TEM studies on the fascicular organization of the human vagus nerve showed laterality with right-sided preference, sexual dimorphism with female dominance, and heterogeneity with marked intra- and inter-individual variability in multiple expressions of cervical fascicle size and numbers. The novel finding of numerous small vagal fascicles, not previously detectable by non-invasive imaging techniques or routine histochemical stains of paraffin-embedded or frozen sections, contributed to the distinction between sides and sexes as well as the pronounced diversity in fascicle morphology for the human vagus nerve. Next, ultrastructural characterization of all myelinated and unmyelinated axons within individual fascicles showed extensive diversity

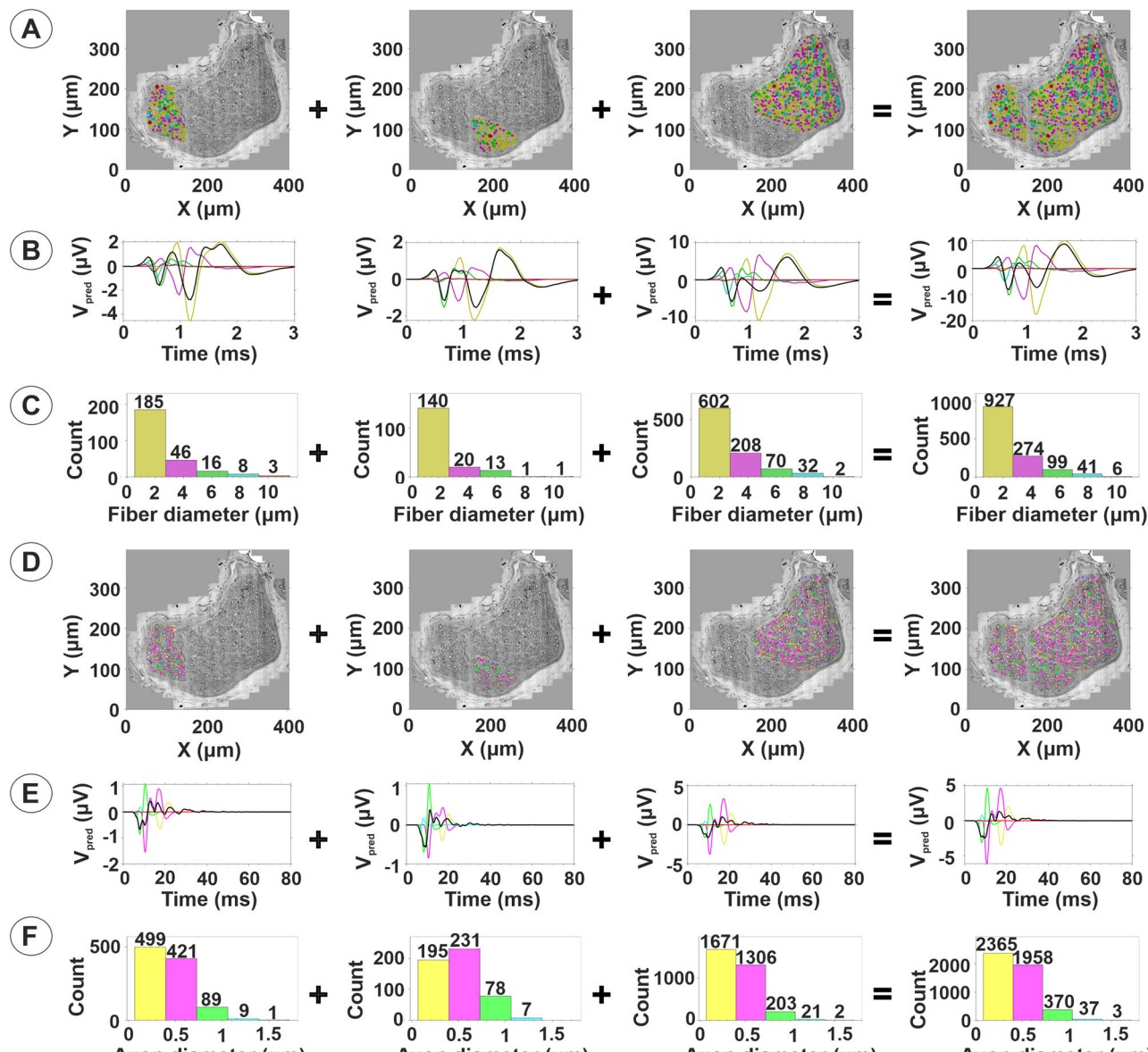

**Fig. 9 | The sum of sub-fascicles is equal to the whole fascicle.** Unique sub-fascicle divisions are observed in the cervical segment of the human vagus nerve and can be shown and characterized with the use of the HAPI tool. HAPI-generated CNAP predictions allow for the exploration of the relative contributions of single fiber action potentials (SFAPs) from each fiber at a particular radial location in the nerve to the CNAP of each individual sub-fascicle and for the whole fascicle. In rows **A** and **D**, columns 1–3 show features of individual sub-fascicles, including TEM montages with superimposed digital segmentation of myelinated fibers (**A**) and unmyelinated fibers (**D**). In rows **B** and **E**, columns 1–3 show the predicted maximal CNAP responses for the myelinated (**B**) and unmyelinated (**E**) fibers in each sub-fascicle. In this simulation, we predicted the maximal expected CNAP response if measured at a conduction distance of 10 mm from the stimulating cathode and using a bipolar Pt/Ir recording cuff electrode with 0.5 mm inner diameter and 2.0 mm inter-electrode spacing. In rows **C** and **F**, columns 1–3 show the fiber diameter distributions of myelinated (**C**) and unmyelinated (**F**) fibers that are expected to produce the CNAP volleys in same column of rows **B** and **E**, respectively. While the fiber size distributions are similar for each sub-fascicle, the numbers of fibers in each bin differ for each fascicle, as do the relative amplitudes of the predicted CNAPs. Marker sizes are scaled according to corrected SAE-diameters and colors are used to map the CNAPs expected from each sub-fascicle to the sizes and locations of all fibers in each sub-fascicle. The rightmost column (column 4) shows the locations, sizes, distribution, and expected CNAPs for the whole fascicle (Myelinated axons: **A**–**C**; Unmyelinated axons: **D**–**F**). We observe that summation of the data across columns 1–3 produces the data in column 4.

of fiber types and inter-fascicle heterogeneity for the total number of nerve fibers and fiber type compositions based on morphological features of axon size and myelination. Additional 2D-distribution analysis of myelinated, unmyelinated, and the total population of fibers in individual fascicles showed pronounced heterogeneity and significant fiber clustering. TEM mapping and segmentation of all myelinated and unmyelinated nerve fibers in individual fascicles also allowed for the use of the HAPI tool to predict action potential shapes and latencies for individual fibers as well as CNAP predictions for all or subsets of nerve fibers within fascicles, hence providing simulated VNS responses to a variety of stimulation algorithm and

validating the functional heterogeneity of the human vagus nerve. Our findings have direct translational research implications for the refinement of VNS, including the design of stimulation electrodes, algorithms and protocols for recruitment or exclusion of select vagal nerve fiber groups during electrical stimulation, and the development of a precision-medicine approach for individualized treatments of conditions.

## Methods
Vagus nerve biopsies were collected intra-operatively from transplant organ donors according to our established protocol[30]. Separate consents were

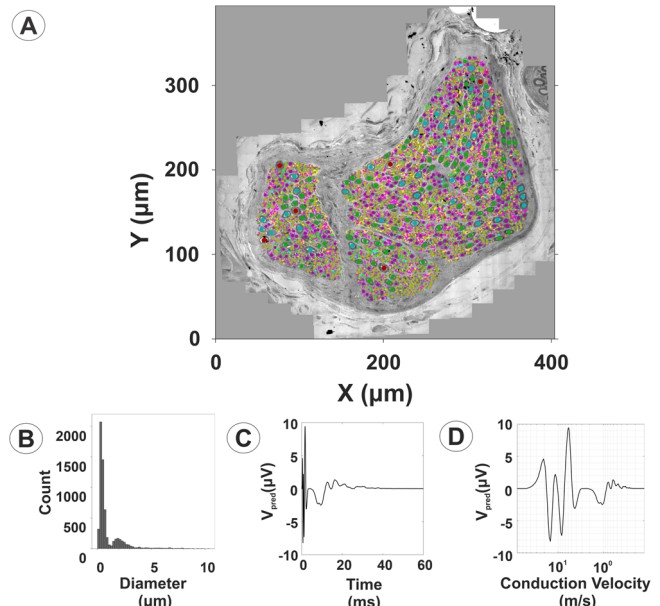

**Fig. 10 | The sum of extracellular single-fiber action potentials from myelinated and unmyelinated fibers produces the maximal CNAP response. A** Combined visualization of myelinated and unmyelinated fibers in all sub-fascicles follows the color scheme and bin range assignments shown in Fig. 9A. Diameter distributions of both myelinated and unmyelinated fibers indicate a wide fiber size range (**B**). The expected maximal CNAP response versus time latency is displayed for the same recording configuration as in Fig. 9 with a conduction distance of 10 mm and use of a bipolar Pt/Ir recording cuff electrode with 0.5 mm inner diameter and 2.0 mm interelectrode spacing (**C**). Plot presentation of the predicted CNAP response versus conduction velocity shows CNAP volleys with conduction speeds in the Aγ, Aδ, B, and C fiber range based on negative peaks and use of the letter system nerve fiber classification (Whitwam, 1976) (**D**). Myelinated and unmyelinated fiber CNAPs possess different relationships that relate conduction speed to axon diameter, myelin thickness, nodal spacing, and other membrane properties. The expected CNAPs are therefore calculated separately by the HAPI tool under the same simulated recording conditions and then summed to produce the expected CNAP for all myelinated and unmyelinated fibers in the fascicle. The HAPI tool can simulate the CNAP from this or other fascicles using virtually any conduction distance or bipolar recording electrode separation, including simulations of single-ended recordings.

obtained for organ donation and the donation of tissues for research studies. A parent and/or legal guardian provided the informed consent for research. No prisoners were included as organ or tissue donors. No photographs were obtained. Prior to the consideration of organ or tissue donation, a diagnosis of brain death was established by independent medical professionals according to the State of Indiana and institutional regulations. All tissue procurement and research procedures were performed in compliance with and approved by the regulatory oversight review committees at the University of Indiana School of Medicine, Indiana Donor Network, and Purdue University. Cervical vagus nerve segments were procured from a continuous series of 27 qualified organ donors. All organ and tissue donors met the diagnostic and medical criteria for organ donation. No cervical vagus nerve tissues from qualified organ donors were excluded from the study. The research team did not participate in the identification and selection of organ donors.

To account for sex as a biological variable, human vagus nerve tissues were procured from both male and female transplant organ donors. Biopsy samples were collected from the distal portion of the cervical vagus nerve of 27 consecutive organ donors and included tissue samples from 14 men and 13 women. Vagus nerve biopsies were collected after the completion of the transplant organ procurement according to our established protocol[30]. In short, distal cervical segments of the vagus nerve, approximately 10–20 mm in length, were collected from the left and right side of the neck.

The vagal nerve segments were immediately placed in a 2% paraformaldehyde +2.5% glutaraldehyde fixative solution at 4 °C. The biopsies were next placed on an oscillating table and underwent immersion fixation over five days at 4 °C with fixative solutions changed daily. After tissue fixation, the vagal segments were rinsed overnight in phosphate buffer at 4 °C, osmicated in 1% osmium tetroxide, rinsed in water, and dehydrated in 30%, 50%, 75%, 95%, and 100% ethanol. The biopsies were next infiltrated in 50% propylene oxide +50% Epon, and embedded in 100% Epon. Transverse sectioning of the plastic resin-embedded vagal nerve biopsies was performed at 0.5 μm thickness with a Histo 45° 8 mm diamond knife (Diatome US®). The semi-thin sections underwent histochemical staining using a 1% toluidine blue solution. The sections were next mounted onto glass slides and cover-slipped. Light microscopic imaging of the cervical vagal biopsy sections was performed using a Nikon E600 light microscope equipped with a Nikon DS-Fi3 camera. The images were captured at 100X and image tiling generated photo-montages of the full cross-section of each cervical vagus nerve segment, including high resolution images of all individual fascicles. The outer and inner perineurium borders for each fascicle was next digitally contoured, using Neurolucida® 360 (MBF Neuroscience), to determine the total fascicle and endoneurium cross-sectional areas, respectively. The shape of individual fascicles was determined by multiple established methods to calculate circularity of nervous structures in morphological studies[31], including the shape factor, form factor, aspect ratio, compactness, and roundness. The shape factor was calculated as $\frac{Perimeter}{\sqrt{Area}}$ (1). The form factor was calculated as $\frac{4\pi Area}{Perimeter^2}$ (2). Aspect ratio was calculated as $\frac{MinFeret}{MaxFeret}$ (3). Compactness was calculated as $\frac{\sqrt{\frac{4}{\pi}Area}}{MaxFeret}$ (4). Roundness was calculated as $\frac{4Area}{\pi MaxFeret^2}$ (5). A circle has a shape factor of 3.54 and a value of 1 for the other four shape outcome measures. MinFeret and MaxFeret refer to minimum and maximum Feret diameters respectively. The LM montages also guided the selection of tissue blocks for subsequent ultrastructural studies of individual nerve fascicles.

For transmission electron microscopy (TEM) studies, plastic resin-embedded cervical vagus nerve tissue blocks were sectioned in the transverse plane at 70–90 nm thickness using an RMC PowerTome Ultramicrotome (Boeckeler Instruments®) and an Ultra 45° 4.0 mm diamond knife (Diatome US®). The ultrathin vagal tissue sections were collected on formvar-coated single slot copper grids and contrasted with uranyl acetate and lead citrate. A Tecnai G2 Spirit Twin TEM (FEI®, ThermoFisher Scientific®) operating at 80 kV was used for the ultrastructural studies. Electron micrographs of cervical vagus nerve fascicles were collected at 6500–42,000 X magnification using a Gatan Orius SC 1000B digital camera (Gatan®, Inc.). Subsequent tiling of the composite TEM images into montages was performed with the use of Adobe Photoshop® (version: 21.1.3 20200508) or Image Composite Editor (ICE®, Microsoft).

Neurolucida® 360 (MBF Bioscience) was used for the TEM data segmentation of all individual myelinated and unmyelinated axons within the endoneurium space of individual nerve fascicles. For myelinated axons, the outer and inner contours of individual myelin sheaths were segmented. Individual segmented nerve fiber enclosures provided raw data files, which included the cross-sectional area, perimeter, and centroid. The shape-adjusted ellipse (SAE) approach was applied to calculate the corrected diameter for myelinated and unmyelinated axons as the minor diameter of an ellipse using the cross-sectional area and perimeter length[31]. The G-ratio was calculated as the ratio of the SAE-corrected diameters for the axon proper and corresponding myelinated fiber[31], and the myelin thickness was calculated as $\left(\frac{fiber\ diameter - axon\ diameter}{2}\right)$ (6).

Predictions and modeling of human vagal compound nerve action potentials (CNAPs) were performed with the aid of a heuristic action potential interpreter (HAPI) tool[57]. HAPI is an emerging tool designed to provide investigators with a common platform for interpreting the type and relative proportion of nerve fibers contributing to individual CNAPs. The HAPI tool was built on early physical and mathematical principles of nerve conduction and superposition, taking into consideration variation in

response latencies and evoked action wave amplitudes to electrical stimulation as well as heterogeneity in nerve fiber size and myelination[58,59]. The HAPI tool allows for the incorporation of all individually segmented nerve fibers in peripheral nerves or nerve fascicles after TEM imaging. Size correction of both myelinated and unmyelinated fibers not sectioned in the perpendicular plane is performed by the HAPI tool using of the SAE approach[31].

The predictions were based on quantitative and size-corrected TEM data on nerve fiber size, axonal size, and myelination for myelinated fibers and axonal size for unmyelinated fibers. The TEM data files were entered into the HAPI software to determine latencies and amplitude peaks of predicted CNAPs for all fibers in individual fascicles or sub-fascicles. Separate CNAP predictions for functional subsets of fibers, based on fiber size and myelination were also performed.

## Statistics and reproducibility

All data are presented as mean ± standard error (SE). The non-parametric ANOVA One-way testing followed by the Kruskal–Wallis test and Dunn's multiple comparison test were performed for comparisons between groups using Prism8® (GraphPad Software, Inc, La Jolla, CA). Pearson's chi-squared test was used to compare observed frequencies between samples. A value of $p < 0.05$ was considered to reflect a statistically significant difference between groups. Frequency distributions for all morphometric size analyses were calculated and plotted using Prism8®. Raw data are included as Supplementary Data 1.

## Data availability

Numerical source data for all graphs in the manuscript can be found in Supplementary Data 1 file.

## Code availability

A GitHub link is provided for the HAPI code used to produce all included CNAP predictions and figures (https://github.com/ybeshay3/Heuristic-Action-Potential-Interpreter-Applied-to-the-Human-Vagus-Nerve-Biscola-et-al.-2024)[60].

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

## Acknowledgements

The studies were supported by the National Institutes of Health (NIH) awards OD026585 (L.A.H.), OD023847 (T.L.P.), DK027627 (T.L.P.), and OD028183 (M.P.W.), and by the Dr. Miriam and Sheldon G. Adelson Medical Research Foundation (L.A.H.).

## Author contributions

Conceptualization, N.P.B., P.M.B., T.L.P., M.P.W., and L.A.H; Methodology, N.P.B., P.M.B., P.V.M., T.L.P., M.P.W., and L.A.H; Software, M.P.W.; Validation, N.P.B., P.M.B., Y.B., M.P.W., and L.A.H; Formal Analysis, N.P.B., P.M.B., Y.B., M.P.W., and L.A.H; Investigation, N.P.B., P.M.B., Y.B., E.S., P.V.M., T.L.P., M.P.W., and L.A.H; Resources, T.L.P., M.P.W., and L.A.H.; Data Curation, N.P.B., M.P.W., and L.A.H.; Writing – Original Draft, L.A.H.; Writing – Review & Editing, N.P.B., P.M.B., Y.B., E.S., P.V.M., T.L.P., M.P.W., and L.A.H.; Visualization, N.P.B., P.M.B., Y.B., M.P.W., and L.A.H.; Funding Acquisition, T.L.P., M.P.W., and L.A.H.

## Competing interests

The authors declare no competing interests.
