## [Transparent Peer Review file · Communications Biology]

Laterality, Sexual Dimorphism, and Human Vagal Projectome Heterogeneity Shape Neuromodulation to Vagus Nerve Stimulation

Corresponding Author: Dr Leif Havton

Version 0:

Reviewer comments:

Reviewer #1

(Remarks to the Author)

Dear Authors

I have read with great interest your paper that I mean to commend for publication. The study is rich in details and the conclusions are confirmed by a robust analysis of histological specimens.

I have three questions: 1) according to your study, there is a right-sided major development of vagus nerve in women: do you think this can lead to some modifications in VNS therapy?; 2) do you think that intraoperative neuromonitoring could help in identifying the most efficacious segment of the nerve to be implanted during VNS surgery?; 3) furthermore, do you believe that VNS stimulation could enhance Vagus Nerve sprouting or modifications?

Sincerely yours

Reviewer #2

(Remarks to the Author)

Reviewer Report

The paper claims novel findings of small fascicles in the vagus nerve trunk not previously detected. Also, individual fascicle showed myelinated and unmyelinated nerve fibers of clear diversity of fiber types and heterogeneity of total number with evidence of laterality and sex differences. These findings could improve Vagal Nerve Stimulation protocol for various therapeutic reasons and could minimize off-target effects of VNS. The paper's findings and conclusions are convincing. This work is of translational interest in the field of therapeutic uses of VNS.

Future research, building on the results of this study, could lead to more effective applications of VNS across various fields. The detailed methodology provided could enable other researchers to replicate the work to a considerable extent. It appears that the authors have employed suitable statistical tools to analyze the research data.

Comments: The following comments are to be addressed by the authors.

1. Line 113 & 114: How organ donors were selected? Please verify.
2. Line 119: "There was no statistical difference between the male and female study groups." This sentence could be included in results section and not in materials and methods section.
3. Line 127: "H₂O should be written as "water".
4. Line 140 & 208: it is not clear to the readers what is meant by shape, form, aspect ratio and roundness. A short description could make all these terms clear to the readers.
5. Line 164: Kurdukar et al, 2021 published an abstract of their work in FASEB journal in a special issue of Experimental Biology meeting 2021. In this work the authors utilized a recently developed compound action potential modeling tool. They have not included this tool in that particular abstract.
How the authors developed HAPI? This should be included in detail in the Method section.
6. Line 219: "Both the total fascicle area and endoneurium area were larger on the left side". It is better to mention the right-side readings to be in line with the readings of right-side circumference (line 217).
7. Line 229: There is an error. "Fascicle area" should be "Endoneurium area".
8. Line 235 & 236: "log₁₀ fascicle area and the ratio of 235 endoneurium area/fascicle area showed a strong correlation" while Fig 2J shows R²=0.50 for Left cervical and R²=0.66 for the left cervical. These values indicate a moderate (not strong) correlation.

9. Line 252: What is G-ratio? It should be defined in the article text not only in the Figure 3.

Dr. Ali Abdul Latif, MB,ChB,FRCPE,PhD
Consultant neurologist & Clinical
Neurophysiologist

Version 1:

Reviewer comments:

Reviewer #1

(Remarks to the Author)

Dear authors

congratulations, the paper is amenable for publication

Bests

Reviewer #2

(Remarks to the Author)

Having reviewed the revised manuscript, I believe the authors have adequately addressed the reviewers' comments and have made the necessary revisions.

The paper now appears to be ready for publication.

Reviewer #1

1) According to your study, there is a right-sided major development of vagus nerve in women: do you think this can lead to some modifications in VNS therapy?

Response: Our finding of a difference in the organization of the vagus nerve between men and women, at the fascicular level, raises the possibility that sex-based variables may contribute to the overall variability in therapeutic responses and off-target effects. We believe that additional studies at the sub-fascicular organization of the human vagus nerve may be needed to fully appreciate potential sex-based differences that may contribute to future modifications in VNS therapy. We anticipate a personalized medicine approach to contribute to the development of VNS strategies to optimize therapeutic effects and improve tolerability of neuromodulation by electrical stimulation. We have added text to the Discussion to provide an expanded perspective on this interesting topic.

2) Do you think that intraoperative neuromonitoring could help in identifying the most efficacious segment of the nerve to be implanted during VNS surgery?

Response: It may be possible in the future to perform intra-operative mapping of evoked responses to vagus nerve stimulation at the time of electrode implantation for VNS for the purpose of identifying the most optimal electrode placement site. However, additional studies are needed to identify the best bio-signals to guide such mapping efforts in the clinical setting. The present studies highlight an extensive heterogeneity in organization of the human vagus nerve between subjects and between the sexes that may need to be considered to refine VNS in therapeutics. We have expanded text in the Discussion to provide additional future directions in VNS therapy, in part influenced by the findings in the present manuscript.

3) Furthermore, do you believe that VNS stimulation could enhance Vagus Nerve sprouting or modifications?

Response: To the best of our knowledge, it is unknown whether VNS may result in sustained structural modifications of the organization of the human vagus nerve because of, for instance, sprouting in the periphery or in the brain stem. Our studies were not designed to address this interesting question. However, it may be possible to address such questions on the possible structural plasticity of the vagus nerve in animal models as examples of reverse research translation.

Reviewer #2

1) Line 113 & 114: How organ donors were selected? Please verify.

Response: The vagus nerve biopsies were obtained from a continuous series of 27 qualified organ donors meeting the diagnostic and medical criteria for organ donation and with consent for research participation in place. The inclusion of organ donors and procedures for the procurement of vagus nerve samples for research use met the criteria of and was approved by regulatory oversight review committees at University of Indiana School of Medicine, Indiana Donor Network, and Purdue University. The research team was not involved in the selection of organ donors. We have expanded the Methods section with additional information with regards to the organ donor selection and provided a reference to our established protocol for donor selection and tissue procurement.

2) Line 119: "There was no statistical difference between the male and female study groups." This sentence could be included in results section and not in materials and methods section.

Response: We agree and have moved the sentence to the Results section.

3) Line 127: "H2O should be written as "water".

Response: Done

4) Line 140 & 208: it is not clear to the readers what is meant by shape, form, aspect ratio and roundness. A short description could make all these terms clear to the readers.

Response: Shape, form, aspect ratio, and roundness refer to different mathematical approaches to determine how close a 2-dimensional space to the shape of a circle. We have added term clarifications to the text, where this shape topic is introduced, as well as a relevant reference to the literature for additional in-depth information and formulas for the calculation of the different shape factors.

5) Line 164: Kurdukar et al, 2021 published an abstract of their work in FASEB journal in a special issue of Experimental Biology meeting 2021. In this work the authors utilized a recently developed compound action potential modeling tool. They have not included this tool in that particular abstract. How the authors developed HAPI? This should be included in detail in the Method section.

Response: We have provided, as suggested, additional text in the Methods with an expanded description of the HAPI tool development. In addition, upon acceptance of the present manuscript for publication in Communications Biology, a GitHub link with the HAPI code used for the CNAP predictions and figures will become available to the public under the Code Availability section of the paper.

6) Line 219: " Both the total fascicle area and endoneurium area were larger on the left side ". It is better to mention the right-side readings to be in line with the readings of right-side circumference (line 217).

Response: We agree and have changed the reference to the right vs. left sided findings for better consistencies.

7) Line 229: There is an error. "Fascicle area" should be "Endoneurium area".

Response: Thank you for catching this error. We have corrected the error.

8) Line 235 & 236: " log10 fascicle area and the ratio of 235 endoneurium area/fascicle area showed a strong correlation " while Fig 2J shows R2=0.50 for Left cervical and R2=0.66 for the left cervical. These values indicate a moderate (not strong) correlation.

Response: We agree and have revised the text to indicate a moderate correlation.

9) Line 252: What is G-ratio? It should be defined in the article text not only in the Figure 3.

Response: A G-ratio is an expression of myelination and is calculated as the diameter of an axon divided by the diameter the same axon with the surrounding myelin sheet included. We have added the definition to the article text.